# Localization Techniques for Non-Palpable Breast Lesions: Current Status, Knowledge Gaps, and Rationale for the MELODY Study (EUBREAST-4/iBRA-NET, NCT 05559411)

**DOI:** 10.3390/cancers15041173

**Published:** 2023-02-12

**Authors:** Maggie Banys-Paluchowski, Thorsten Kühn, Yazan Masannat, Isabel Rubio, Jana de Boniface, Nina Ditsch, Güldeniz Karadeniz Cakmak, Andreas Karakatsanis, Rajiv Dave, Markus Hahn, Shelley Potter, Ashutosh Kothari, Oreste Davide Gentilini, Bahadir M. Gulluoglu, Michael Patrick Lux, Marjolein Smidt, Walter Paul Weber, Bilge Aktas Sezen, Natalia Krawczyk, Steffi Hartmann, Rosa Di Micco, Sarah Nietz, Francois Malherbe, Neslihan Cabioglu, Nuh Zafer Canturk, Maria Luisa Gasparri, Dawid Murawa, James Harvey

**Affiliations:** 1Department of Gynecology and Obstetrics, University Hospital Schleswig-Holstein, Campus Lübeck, 23538 Lübeck, Germany; 2Department of Gynecology and Obstetrics, Die Filderklinik, 70794 Filderstadt, Germany; 3Aberdeen Breast Unit, Aberdeen Royal Infirmary, Aberdeen AB25 2ZN, UK; 4Breast Surgical Oncology, Clinica Universidad de Navarra, 28027 Madrid, Spain; 5Department of Molecular Medicine and Surgery, Karolinska Institutet, 17177 Stockholm, Sweden; 6Department of Surgery, Capio St. Göran’s Hospital, 11219 Stockholm, Sweden; 7Breast Cancer Center, University Hospital Augsburg, 86156 Augsburg, Germany; 8Breast and Endocrine Unit, General Surgery Department, Zonguldak BEUN The School of Medicine, Kozlu/Zonguldak 67600, Turkey; 9Department for Surgical Sciences, Faculty of Pharmacy and Medicine, Uppsala University, 75236 Uppsala, Sweden; 10Section for Breast Surgery, Department of Surgery, Uppsala University Hospital, 75236 Uppsala, Sweden; 11Nightingale & Genesis Breast Cancer Prevention Centre, Manchester University NHS Foundation Trust, Faculty of Biology, Medicine and Health, University of Manchester, Manchester M13 9PL, UK; 12Department for Women’s Health, University of Tübingen, 72076 Tübingen, Germany; 13Bristol Medical School (THS), Bristol Population Health Science Institute, Bristol BS8 1QU, UK; 14Guy’s & St Thomas NHS Foundation Trust, Kings College, London SE1 9RT, UK; 15Department of Breast Surgery, San Raffaele University and Research Hospital, 20132 Milan, Italy; 16Department of Surgery, Breast Surgery Unit, Marmara University School of Medicine and SENATURK Turkish Academy of Senology, Istanbul 34854, Turkey; 17Department of Gynecology and Obstetrics, St. Louise Frauen-und Kinderklinik, 33098 Paderborn, Germany; 18Department of Surgical Oncology, Maastricht University Medical Center, 6229 HX Maastricht, The Netherlands; 19Division of Breast Surgery, Department of Surgery, Basel University Hospital, 4031 Basel, Switzerland; 20European Breast Cancer Research Association of Surgical Trialists (EUBREAST), 73730 Esslingen, Germany; 21Department of Gynecology and Obstetrics, Heinrich-Heine-University Düsseldorf, 40225 Düsseldorf, Germany; 22Department of Gynecology and Obstetrics, University Hospital Rostock, 18059 Rostock, Germany; 23Department of Surgery, Faculty of Health Sciences, University of the Witwatersrand, Johannesburg 2000, South Africa; 24Breast and Endocrine Surgery Unit, Groote Schuur Hospital, University of Cape Town, Cape Town 7935, South Africa; 25Istanbul Faculty of Medicine, Department of General Surgery, Istanbul University, Istanbul 34093, Turkey; 26Department of General Surgery, Kocaeli University School of Medicine, Kocaeli 41001, Turkey; 27Department of Gynecology and Obstetrics, Ospedale Regionale di Lugano EOC, 6900 Lugano, Switzerland; 28Centro di Senologia della Svizzera Italiana (CSSI), Ente Ospedaliero Cantonale, Via Pietro Capelli 1, 6900 Lugano, Switzerland; 29Faculty of Biomedical Sciences, Università della Svizzera Italiana (USI), Via Giuseppe Buffi 13, 6900 Lugano, Switzerland; 30General Surgery and Surgical Oncology Department, Collegium Medicum, University in Zielona Gora, 65-417 Zielona Góra, Poland; 31Nightingale & Genesis Breast Cancer Prevention Centre, University Hospital of South Manchester NHS Foundation Trust, Manchester M13 9PL, UK

**Keywords:** breast cancer, localization technique, non-palpable lesion, intraoperative ultrasound, wire-guided localization, magnetic seed, radioactive seed, radar reflector, radiofrequency identification tag

## Abstract

**Simple Summary:**

Most breast cancers are small and can be treated using breast-conserving surgery. Since these tumors are non-palpable, they require a localization step that helps the surgeon to decide which tissue needs to be removed. The oldest localization technique is a guidewire placed into the tumor before surgery, usually using ultrasound or mammography. Afterwards, the surgeon removes the tissue around the wire tip. However, this technique has several disadvantages: It can cause the patient discomfort, requires a radiologist or another professional specialized in breast diagnostics to perform the procedure shortly before surgery, and 15–20% of patients need a second surgery to completely remove the tumor. Therefore, new techniques have been developed but most of them have not yet been examined in large, prospective, multicenter studies. In this review, we discuss all available techniques and present the MELODY study that will investigate their safety, with a focus on patient, surgeon, and radiologist preference.

**Abstract:**

Background: Surgical excision of a non-palpable breast lesion requires a localization step. Among available techniques, wire-guided localization (WGL) is most commonly used. Other techniques (radioactive, magnetic, radar or radiofrequency-based, and intraoperative ultrasound) have been developed in the last two decades with the aim of improving outcomes and logistics. Methods: We performed a systematic review on localization techniques for non-palpable breast cancer. Results: For most techniques, oncological outcomes such as lesion identification and clear margin rate seem either comparable with or better than for WGL, but evidence is limited to small cohort studies for some of the devices. Intraoperative ultrasound is associated with significantly higher negative margin rates in meta-analyses of randomized clinical trials (RCTs). Radioactive techniques were studied in several RCTs and are non-inferior to WGL. Smaller studies show higher patient preference towards wire-free localization, but little is known about surgeons’ and radiologists’ attitudes towards these techniques. Conclusions: Large studies with an additional focus on patient, surgeon, and radiologist preference are necessary. This review aims to present the rationale for the MELODY (NCT05559411) study and to enable standardization of outcome measures for future studies.

## 1. Introduction

Surgical excision of a non-palpable breast lesion requires some form of breast localization device. Despite multiple available solutions, a majority of units use wire-guided localization (WGL) due to the high efficacy and low cost [1,2]. Other techniques, e.g., radioactive seed localization, radio-occult lesion localization (ROLL), and intraoperative ultrasound, have become established in a smaller number of centers but have not gained widespread adoption. While WGL has clear benefits in terms of cost, efficacy, and a trained workforce, it also carries several weaknesses, including logistical difficulties due to the need of placement on the day of surgery and the potential for displacement. Despite widespread WGL use, a majority of breast surgeons have voiced a preference to switch to an alternative technique [2]. Since 2016, a new generation of localization devices has entered the market including SAVI SCOUT^®^, LOCalizer™, Magseed^®^, Pintuition^®^, EnVisio^®^, and Molli™ (Figure 1). The IDEAL framework provides a system for evaluating surgical innovations from “first in human” (stage 1), “exploration” (stage 2), and “assessment” (stage 3) to “long term study” (stage 4) [3]. Most novel techniques are moving through from a development stage into an exploratory phase, where they are becoming more standardized and replicated by others. Acknowledgement of learning curves is important [4,5].

The European Breast Cancer Research Association of Surgical Trialists (EUBREAST) and the iBRA-NET have initiated the MELODY (Methods for Localization of Different types of breast lesions) study to assess breast localization techniques and devices from several perspectives. MELODY is a multinational prospective intergroup cohort study which enrolls breast cancer patients undergoing breast-conserving surgery using imaging-guided localization. As an IDEAL stage 2b/3 observational study, it aims to explore the safety, efficacy. and patient-/clinician-reported outcomes of different localization techniques [6]. The study is designed to ensure thorough surgical evaluation and yield high-quality evidence for both patients and clinicians, potentially allowing evidence-based adoption of these techniques by national bodies and regulatory authorities.

This narrative review aims to identify the current knowledge base of established and newer localization techniques, to help inform the MELODY (NCT05559411) study design and to enable standardisation of outcome measures for future studies.

## 2. Current Evidence of Different Localization Techniques

### 2.1. Wire-Guided Localization (WGL)

For decades, WGL was the main localization technique, and is still considered the gold standard in many countries [7,8]. Initially developed in the 1960s and popularized in the 1970s and 1980s, the technique involves a wire or a needle placed preoperatively into the lesion under sonographic or mammographic guidance, usually followed by ultrasound or radiography of the subsequently surgically removed specimen (Figure 2) [9]. Disadvantages of WGL, such as the necessity to perform the procedure on the day of surgery or—less frequently—on the day before, the possibility of wire dislocation, and patient discomfort and distress, have led to a search for alternative strategies.

A recent analysis from the Netherlands including 28,370 patients showed that probe-guided localization is replacing WGL, with the use of radioactive seed localization having increased from 16% to 61% between 2013 and 2018, while WGL decreased from 75% to 32% [1].

To date, all randomized controlled trials (RCTs) on newer localization techniques have compared them to the WGL (Table 1, Table 2) [8,10,11]. The positive margin rate of WGL was reported to be in the range of 15–21% [8,10,12,13]. Two network meta-analyses of RCTs showed that margin positivity and reoperation rates of all techniques were similar, except for intraoperative ultrasound that led to significantly reduced margin positivity and re-excision rates [10,11].

### 2.2. Radioactive Localization

Radio-guided surgery is a wire-free approach to assist surgical excision of non-palpable breast lesions by using a gamma probe to detect a preinserted marker. Two forms of radioactive localization are currently in use: radioactive seed localization (RSL) is based on the detection of a small 125-iodine seed, while radioactive occult lesion localization (ROLL) relies on the identification of preinjected radiocolloid (99m Technetium) [9].

Radioactive seed localization was first described in 1999 in a pilot study that included 25 patients who underwent excisional biopsy [27]. The seed is composed of titanium containing 3.7 to 10.7 MBq ^125^I (iodine) with a half-life of 60 days. Seeds are introduced via a needle under sonographic or mammographic guidance directed into the index lesion, and appropriate insertion is confirmed via subsequent imaging. Due to the long half-life, it is possible to insert the seed weeks or even months before the surgical intervention, making its use also an option in the neoadjuvant setting. During surgery, the seed is detected by a standard intraoperative handheld gamma probe, and the area of greatest activity projecting directly over the lesion is easily located to allow the most appropriate incision to be placed. In some countries, such as the United States, Canada, and the Netherlands, RSL is considered a standard approach [1]. Beyond localization of breast lesions, there is an increasing body of evidence for marking axillary lymph nodes with radioactive seeds [28,29].

RSL is one of the best validated wire-free localization methods. It has been investigated in several RCTs and meta-analyses [9,10,11]. A Cochrane review published in 2015 concluded that RSL was equally reliable compared with WGL, but the authors stressed the need for further, fully powered RCTs. Since then, more RCTs were published [30,31,32]. The successful excision rate, defined as removal of the index lesion with clear margins, was reported in the range of 99.4–100% [9,30,31,33,34]. In the available studies, the failure rate was comparable to that of WGL.

In the RCTs comparing RSL with WGL, the rate of positive margins was generally lower for RSL [30,31,32,33,34,35]. However, a recent network meta-analysis of RCTs evaluating optimal localization strategies for non-palpable breast cancers, including 24 studies, suggested no significant differences when comparing RSL with WGL for both margin positivity (OR: 0.677, 95% CI 0.397–1.110) and reoperation rates (OR: 0.685, 95% CI 0.341–1.260) [10]. In contrast, another meta-analysis comparing RSL with WGL, including both retrospective and prospective studies, outlined that RSL was superior to WGL by providing negative margins (RR: 0.72, 95% CI 0.56–0.92, *p* = 0.01) and lower reoperation rates (RR: 0.68, 95% CI 0.52–0.88, *p* = 0.004) [36,37].

While RSL is a popular localization method in some countries, the seeds are not approved for such use in others. Due to complex radiation safety regulations, the use of iodine seeds requires trained personnel, the implementation of standard operating procedures, and, depending on the country, a formal submission to a radiation protection agency for authorization. It may be mandatory to provide a facility diagram and description of the location(s) where the radioactive sources will be received, used, and stored. Each seed must be accounted for, and, unlike other localization devices, the loss of a seed is considered a serious breach of radiation safety. For this reason, seeds are generally implanted under ultrasound or mammographic, but not MRI, guidance. The MRI safety concern is related to the possibility of losing a seed in the MRI scan room without the option of using a hand-held Geiger counter to locate the seed [38].

Several studies analyzed the cost-effectiveness of RSL. The necessity to adhere to strict radiation safety regulations results in substantial upfront costs of RSL implementation [39]. The estimated costs per patient vary strongly between studies; while some reported slightly higher costs for RSL than for WGL (EUR 2834 vs. EUR 2,617 per patient, respectively) [39], others showed a lower average cost per patient for RSL (USD 251 compared to USD 1130 for WGL) [40]. Possibly, the cost-effectiveness of RSL depends on the health-care payment system (fee-for-service vs. bundled) [41].

Regarding MRI compatibility after placement, radioactive seeds may cause minimal and usually not clinically relevant susceptibility artifacts, similar to those observed around clips/coils [42]. Migration of implanted seeds seems rare, and was reported as 0.9 mm on average [42]. Although some early studies reported lower specimen volumes in patients receiving RSL [34], the available meta-analyses show no significant differences regarding specimen size, weight, or volume between patients undergoing RSL and WGL [10,11]. Few studies analyzed patient satisfaction with the localization procedure. In a RCT by Bloomquist et al., significantly fewer patients in the RSL arm reported moderate to severe pain during the localization procedure compared to the WGL arm, and the overall convenience of the procedure was rated as very good to excellent in 85% of RSL patients compared to 44% of WGL patients (*p* < 0.0001) [31]. No randomized data are available on surgeon or radiologist satisfaction with the technique.

The Radio-guided Occult Lesion Localization (ROLL) technique was primarily introduced by the team at the European Institute of Oncology in Milan in 1999 [43]. This procedure uses 99m Technetium-labelled colloidal human serum albumin as a radioactive tracer to label the lesion under sonographic or mammographic guidance. Similar to RSL, the tracer is localized using a handheld gamma probe and can be used for simultaneous sentinel node biopsy. The combined procedure is commonly referred to as SNOLL (Sentinel Node plus Occult Lesion Localization) [44]. The gamma radiation dose to the patient and the operators is very low and well within safe radiation regulatory limits.

Several RCTs and meta-analyses have examined the use of ROLL. A Cochrane review showed comparable rates of successful excision of the target lesion between the technique and WGL [9]. In the RCTs comparing WGL with ROLL, the rate of positive margins was reported to be higher in the WGL arm, but the differences were mostly not statistically significant [45,46,47,48,49,50,51,52,53,54]. In a recent network meta-analysis of RCTs evaluating optimal localization strategies for non-palpable breast cancers, including 24 studies, margin positivity rate was 20.1% for WGL and 17.2% for ROLL [10].

While ROLL is a popular technique in some parts of the world (Turkey, Australia, Latin America), it remains unknown in others. In clinical practice, the main disadvantage of ROLL is the necessity of the injection on the day of surgery or the day before surgery, which may be associated with difficulties in synchronizing the schedules of the nuclear medicine, radiology, and the operating room. Further, strictly seen, 99m Technetium is approved for sentinel lymph node identification, and not lesion localization, so there might be some concern regarding a potential off-label use in some countries.

The cost-effectiveness of ROLL has not been evaluated in large RCTs. In two RCTs comparing costs, ROLL (mean cost: EUR 182) was found to be slightly more expensive than WGL (mean cost: EUR 163) [10]. The technique is MRI compatible: It does not cause MRI artifacts and allows localization of lesions observed only on MRI [55]. Localization failures are rare [9]. Regarding specimen size, weight, and volumes, two recent meta-analyses reported no significant differences compared to WGL [10,11]. However, in the largest RCT, ROLL led to the excision of larger volumes [51].

Surgeon satisfaction rate was highest (98.4%) for ROLL when compared to the rate of 66% for WGL [10]. Conflicting results were reported with regard to patient pain score during the localization procedure [50,51]. No significant differences were found on patient-reported cosmetic results and pain between ROLL and WGL six months after surgery [51].

### 2.3. Magnetic and Paramagnetic Localization

Moving further from WGL, and in order to address the strict regulatory issues with regards to access, availability, handling, and disposal of radioactive material, several markers based on the principle of magnetic detection have been developed in recent years. The perceived advantage in such a device is that it allows for wire-free and radiation-free localization. Additionally, it yields the potential to facilitate logistics of localization, as it can be implanted many days before surgery. At present, both magnetic and paramagnetic markers are available for clinical use. Paramagnetic markers have a small susceptibility to magnetic fields and become temporarily magnetized in a presence of an externally applied magnetic field, while magnetic markers are permanent magnets. Metallic instruments may interfere with the detection of magnetic and paramagnetic markers, and both types of markers lead to significant MRI artifacts, limiting its use in the neoadjuvant setting [56].

The most well-studied marker in this category is a 5 × 1 mm long, steel paramagnetic marker (Magseed, Endomag, Cambridge, UK) investigated in multiple cohort studies (Figure 3) [12,57,58]. This device is licensed for both breast and axillary placement, and early studies demonstrated no migration within the breast [59]. In a recent multi-center study from the UK iBRA-NET, a total of 946 Magseed-guided excisions were compared with 1170 wire-guided excisions [12]. The authors found that the use of Magseed resulted in more successful index lesion removal (99.8% vs. 99.1%, *p* = 0.048) and fewer failed localizations (1.64% vs. 1.98%, *p* = 0.032). While it was associated with less risk of dislocation (0.4% vs. 1.4%, *p* = 0.039), the secondary outcomes (minimum margins, specimen sizes, re-excision surgery, postoperative complications) were comparable. In terms of logistics, Magseed-guided surgery had an earlier start on the day of surgery. Previous reports from the UK had shown similar results; Zacharioudakis et al. demonstrated comparable outcomes between the two techniques (n = 100 patients each arm) with regards to successful identification and removal, margin status, specimen size, and tumor-to-specimen volume ratio [58]. Micha et al. found that re-excision rates were similar in an institutional cohort study comparing Magseed (n = 100) to WGL (n = 100). The use of Magseed did not only achieve smaller specimens but also resulted in higher patient and physician satisfaction, and thus a preference for the magnetic technique [60]. Magseed localization is compromised by metal instruments and can be challenging when the seed is placed deep in the breast [61]. There is no evidence for superior cost-effectiveness or patient-reported outcomes when comparing it to other localization devices [62].

### 2.4. Sirius Pintuition

The Magnetic Marker Localization (MaMaLoc) is a permanent magnetic marker that has been developed for breast localization. This marker has evolved with its own detection system; it is commercially available as Sirius Pintuition. The probe used for detection has an additional tool to show not only the distance to the seed, but the angle as well (Figure 4). Available data at the time of writing of this manuscript are so far limited to institutional reports presented as congress abstracts [63,64]. The originally developed device, the MaMaLoc, was compared to WGL in a small RCT (n = 70), powered to detect differences in the System Usability Scale (SUS) [15]. In this trial, all markers could be successfully retrieved. The positive margin rate was significantly lower in the magnetic marker arm (8% vs. 18% in the WGL group), but reoperation rates were similar (4% vs. 6%, respectively).

Sirius Pintuition is approved in the EU for placement for up to 180 days in any soft tissue, allowing for use in both breast and axilla. There is little evidence base to establish its migration rate, effectiveness, failure rate, cost-effectiveness, complication rates, or patient/physician satisfaction. The performance and safety of the Pintuition device is currently undergoing evaluation in a UK multi-center comparative cohort study [65].

The detection probe is compatible with standard metal instruments, as long as they are not magnetized since this may lead to interference with the probe. Thus, it might be prudent to have one set of non-metallic instruments available. The main disadvantage of all magnetic markers is the creation of 5–6 cm artifacts surrounding the marker when using MRI. Therefore, if tumor response is to be assessed by MRI, magnetic markers should not be placed in the vicinity of the tumor area before neoadjuvant chemotherapy (NACT) [66].

While no such data are available regarding Sirius Pintuition localization, the abovementioned RCT on MaMaLoc vs. WGL showed comparable specimen weight and volume in both arms [15]. In this trial, patients reported more discomfort and pain during guidewire placement, but this result may be biased since patients allocated to WGL did not receive local anaesthesia whereas those allocated to the MaMaLoc did. Patients’ overall satisfaction with the localization technique was rated significantly better for MaMaLoc than for WGL. Similarly, MaMaLoc localization led to higher surgeon satisfaction scores measured by a procedure-specific questionnaire, and surgeons would have preferred the MaMaLoc technique in 56% of cases. No preference was reported in 38% of cases, and WGL was preferred in only 7%.

The Magnetic Occult Lesion Localization Instrument (MOLLI) is another magnetic (not paramagnetic) marker with its own probe-based detection system. The current evidence is very limited and stems from only one feasibility study (n = 20) where all patients received a radioactive seed together with the MOLLI [16]. In this study, retrieval of the MOLLI was successful in all cases and with high physician satisfaction, but the small population studied, and study design do not allow for more robust conclusions. Finally, another magnetic marker has been developed in Japan: the Guiding-Marker System^®^, which is compatible with the handheld TAKUMI magnetic probe. The system has been validated in a single-arm multicenter study (n = 87), where marker retrieval was 100% and the re-excision rate was 6.1% [17].

In conclusion, magnetic guidance for tumor localization seems a promising technique, with a variety of devices that are commercially available. However, all evidence stems from non-randomized data, the only exception being a small RCT on MaMaLoc [15]. At the time of writing, a phase 3, pragmatic multicenter randomized controlled trial (MagTotal) is accruing data comparing Magseed and WGL (ISRCTN11914537). Given the differences of available devices in principle (paramagnetic vs. magnetic), probe compatibility, possibilities to utilize as a single platform for breast and axillary surgery, and the imbalance among them in terms of published data, further evaluation is needed.

### 2.5. Radar Reflector Localization

The SAVI SCOUT is a zero-radiation breast localization and surgical guidance system using micro-impulse radar technology for the removal of non-palpable breast lesions. It was introduced in 2015 and is approved by FDA and CE for long-term placement in breast, lymph nodes, and soft tissue. The reflector is activated by infrared light impulses generated by the console probe and uses two antennas to reflect an electromagnetic wave signal back to the handpiece. It can be placed using ultrasound or stereotactic guidance (Figure 5).

Initial successful data from a pilot study led to a multicenter study [67]. The primary endpoints were the rates of successful reflector placement, localization, and removal in a patient cohort scheduled to have an excisional biopsy or breast-conserving surgery of a non-palpable breast lesion. SCOUT reflectors were successfully placed in 153 of 154 patients, but in one case, the reflector was placed at such a distance from the target that an additional wire had to be placed. All 154 lesions and reflectors were successfully removed during surgery [67].

A systematic review and pooled analysis of 842 cases (11 studies) revealed an overall successful deployment rate of 99.64% and a successful retrieval rate of 99.64% using the radar reflector system. A statistically significant difference in re-excision rate was found in a smaller pooled analysis conducted across four studies comparing radar reflectors and WGL (12.9% and 21.1% respectively, *p* < 0.01) [4]. This should be interpreted with caution as each study was small, two of these studies are unpublished, and only 264 patients were included in this analysis.

The migration rate of the SCOUT reflector post-placement is low at 1.3%, and location stability was demonstrated across multiple studies up to 516 days post placement [4,67,68,69,70,71]. MRI artifacts may occur but are smaller than those created by magnetic or RFID markers [56,68]. There is no significant evidence evaluating the size of the surgical specimen or cost-effectiveness of the device. There is a failure rate of the device through damage of the antennae prior to surgery or by diathermy, but its magnitude and clinical impact are unclear. There is evidence demonstrating good patient, physician, and radiologist satisfaction but this is limited to a single-arm study [58].

### 2.6. Radiofrequency Identification Tags

Radiofrequency identification (RFID; LOCalizer, Hologic Inc., Santa Clara, CA, USA) is a relatively new but promising technology. The LOCalizer received FDA approval in April 2017 and European CE marking in October 2018, and is approved for marking of breast lesions, not axillary nodes. The RFID marker is a small radiofrequency ‘tag,’ identified with a small portable hand-held device which also comes with a pencil-sized single use probe (Figure 6). It displays the real-time distance to the tag in millimeters, and a unique tag identification number discerns each individual tag if more than one was inserted. [72].

The implantable seed is large compared to other markers, 11 × 2 mm, requiring a 12G needle for deployment. Since the needle is relatively blunt and does not penetrate the skin easily, a small skin incision is needed [73]. Hypothetically, the needle size may cause a wider tract that can result in seed migration. While this had not been reported in the limited literature, there are descriptions that the tag may move intraoperatively while the specimen is being retracted and mobilized [74,75]. Some also have reported loss of previously placed titanium marker clips while inserting the tags [73]. Moreover, the large needle size may pose a challenge to accurate insertion in the dense breast and in hard masses where the tag sometimes resides at the edge of the lesion [73].

An important consideration when planning for RFID are its potential interference with defibrillators and pacemakers, so RFID should be avoided in these patients [74]. Furthermore, a significant MRI artifact of about 2–2.5 cm needs to be accommodated for and is partly caused by the glass encasing [18,74]. Deep lesions in larger breasts can pose a challenge as the RFID detection range is 6 cm [72]. In available studies, patients felt that the procedure went smooth and was easier than expected, with high patient satisfaction rates [76,77], while surgeons and radiologists reported that the device was at least as fast and reliable as WGL [76] or even better [77].

The published body of evidence is limited but growing. In a recent systematic review, nine prospective and retrospective studies were included. Seven studies including 1151 patients and 1344 tags showed a pooled accurate deployment rate of 99.1%, a retrieval rate of 100%, and a re-excision rate of 13.9%. This suggests the device may not migrate although this had not been specifically investigated. Two further studies compared RFID with WGL; the pooled re-excision rate was comparable at 15.6% (20/128 vs. 44/282, respectively, *p* = 0.995) but the datasets are relatively small [18]. Furthermore there are no comparative data regarding patient, surgeon or radiologist experience, cost-effectiveness, or size of surgical specimens [78]. Most data stem from single-center, heterogeneously designed studies at risk of bias, which underlines a need for high-quality data collection to validate early, promising datasets. Although LOCalizer is only licenced for use in the breast, some have also used it to mark axillary nodes for targeted dissection [5].

### 2.7. Intraoperative Ultrasound

In the first publication on intraoperative ultrasound (IOUS)-guided surgery in 1988, Schwartz and colleagues found that ultrasound (US) was an accurate and effective tool for localizing breast masses, thus facilitating the surgical excision [79]. Since then, multiple manuscripts have reported on the use of IOUS to guide breast-conservative surgery in non-palpable breast cancer [80,81,82]. Using this technique, no preoperative localization procedure is necessary. IOUS is performed using a multifrequency probe covered in sterile sheath that ranges from 7 Mhz to 18 Mhz. Smaller probes that are easily introduced into the breast incision can be incorporated to improve visibility during surgery. The method is limited to targets visible on US (either the lesion itself or a sonographically visible marker (Figure 7)) [83]. Furthermore, an US machine needs to be available in the operating room during the procedure, and surgeon training in breast ultrasound is a requirement. A major reported benefit of IOUS is the omission of preoperative localization, which avoids the burden of an additional radiology appointment and facilitates an easy workflow towards surgery. IOUS also allows for continuous margin assessment during surgery and ex vivo margin evaluation directly after specimen removal.

The available evidence on IOUS stems from several RCTs and meta-analyses, as well as cohort studies [8,10,11]. Three RCTs compared IOUS with WGL in non-palpable breast cancer, and a further three RCTs compared IOUS with palpation-guided surgery in patients with palpable tumors [8]. The studies showed a high successful excision rate of target lesions. In addition, various meta-analyses have demonstrated that IOUS significantly increases negative margin rates when compared to WGL [8,10,11,84]. Re-excision of positive or very close margins already identified by intraoperative US reduces the need for a second surgical procedure [8,10,11]. Based on these results, the AGO Breast Committee updated its guidelines in 2022 and endorses IOUS for removal of non-palpable breast cancer with a strong level of recommendation [7,85].

There are few cost-efficiency studies comparing IOUS and WGL, probably due to the complexity considering not only direct but also indirect costs, and their equivalence in quality-adjusted life years (QALY). Available studies show lower costs with IOUS than with WGL [86]. One study evaluated costs related to the use of US-visible clips compared with traditional clips and favored US as a means of localization when feasible. There was an estimated cost saving of USD 36,000 over the 3-year study period despite the initially higher cost of US-visible clips. US localization with US-visible markers thus appears to be cost-effective and spares patients an additional wire placement, which can evoke unnecessary stress and anxiety before surgery [87].

Another advantage of IOUS is the potential for resecting less surrounding healthy breast tissue. The randomized COBALT trial showed lower excised volumes when using IOUS when compared to palpation-guided surgery, which significantly affected cosmetic outcomes and patient satisfaction [88]. No data on surgeon satisfaction with IOUS are available. In the neoadjuvant chemotherapy (NACT) setting, where WGL traditionally has been the standard, IOUS can be used if a residual lesion or an US-visible marker is present [89,90].

Several researchers evaluated the extent of the learning curve to acquire the necessary skills for IOUS. Most surgeons reached mastering level after 7–17 cases, with an average of 11 cases [86]. Others have measured proficiency by observational studies that recorded calculated resection ratios by three surgeons performing ten cases of IOUS surgery each and found this case number to be sufficient to master the technique [91].

### 2.8. Carbon Suspension

The use of a sterile aqueous suspension of carbon powder for the stereotactic marking of occult breast lesions was first described in 56 patients by radiologist Gunilla Svane at Karolinska University Hospital in Sweden in 1983 [25]. The tip of the injection needle was placed in the direct vicinity of the lesion, and a technique was devised allowing the even distribution of carbon suspension over the entire length of a carbon track from lesion to skin, marking the point of entry with a small skin tattoo (Figure 8). Four lesions were missed at first operation or incompletely excised, probably owing to the fact that the concentration of the carbon solution was lower than later recommended in three cases; the fourth case was a fibroadenoma displaced by 5 mm during marking. Subsequently, the method was reported in a few publications [20,92,93,94]. Interruption of the carbon track between skin and lesion may occur during release of pressure after mammography if carbon is placed by stereotaxis, which makes following the carbon track more difficult than when carbon is placed by ultrasound guidance [20]. Since carbon does not yield any acoustic signal, a carbon track placed by stereotaxis entering the skin distant from the lesion location may be challenging, and US-guided placement may facilitate correct excision significantly. In contrast to ink marking, carbon does not bleed into surrounding tissue and does not migrate over time, thus making the method feasible for use before NACT. As carbon is not visible on specimen radiography, it may be combined with clip placement in neoadjuvant cases where the original lesion may undergo complete regression and thus otherwise lose visibility on imaging. The main perceived advantages of carbon localization are its low cost, easy availability, simple logistics, and durability over time, although there is poor quality data supporting its use and no comparative datasets. Currently, this remains a technique that is yet to gain widespread adoption in breast localization and offers no high-quality evidence on accuracy, margin involvement, cost-effectiveness, or patient/surgeon satisfaction.

In contrast, there is a rapidly emerging use of carbon marking for axillary lymph nodes in patients receiving NACT, demonstrating 82–98% accuracy of removal of the targeted node [95,96,97,98,99,100].

## 3. The MELODY Study

MELODY, initiated as an intergroup study between EUBREAST and iBRA-NET, is a prospective non-interventional multicohort study aiming to evaluate different localization techniques for non-palpable breast cancer (http://melody.eubreast.com; accessed on 11 December 2022 (Figure 9)). With a target accrual of 7416 patients, the study is powered to resolve several knowledge gaps. Patients with invasive breast cancer or ductal carcinoma in situ (DCIS), confirmed by minimally invasive biopsy, and scheduled to receive breast-conserving surgery, can be enrolled. The use of NACT and preoperative endocrine therapy are allowed. Marking and localization procedures and treatment modalities are chosen at the discretion of the treating physicians and according to national and institutional guidelines. Inclusion and exclusion criteria are presented in Table 3. Patients will be followed for 30 days postoperatively for potential complications. No long-term surveillance is required.

MELODY is supported by the AGO-B study group, the Oncoplastic Breast Consortium (OPBC), SENATURK, AWOgyn (German Working Group for Reconstructive Surgery in Oncology-Gynecology), and German Breast Group (GBG).

Primary Study Endpoints:Intended target lesion and/or marker removal, independent of margin status on final histopathology;Negative resection margin rates (defined as lesion removal with no invasive or non-invasive carcinoma on ink) at first surgery.

Secondary Study Endpoints:3.Rates of second surgery;4.Rates of secondary mastectomy;5.Resection ratio, defined as actual resection volume divided by the calculated optimum specimen volume;6.Duration of surgery in BC patients, defined as time between first incision and end of skin closure (patients receiving simultaneous reconstructive, oncoplastic or contralateral surgery will be excluded from this analysis);7.Marker dislocation rates;8.Rates of marker placement failure, i.e., marker dislocation requiring a placement of a second marker;9.Rates of localization failure, i.e., failed removal of marker or lesion, or necessity to switch to another intraoperative localization method;10.Patient-reported outcomes (e.g., patient discomfort, pain level, and impairment of breathing);11.Diagnostician/radiologist satisfaction with marking technique;12.Surgeon satisfaction with localization technique;13.Rates of “lost markers” (defined as markers placed prior to surgery and not retrieved at surgery);14.Volume and weight of resected tissue;15.Impact of experience of study sites on other outcome measures, depending on the localization technique used;16.Impact of self-reported ethnicity on outcome measures;17.Evaluation of surgical standards of care in different countries;18.Evaluation of economic resources required for different localization techniques (material costs, operative time etc.);19.Evaluation of MRI artifacts;20.Evaluation of complication rates related to marker placement;21.Evaluation of peri- and postoperative complication rates.

The first MELODY study site has opened in Q4 2022. Currently, 20 countries are planning to participate in the study, most of which are in the process of applying for ethical approval.

## 4. Conclusions

Wire-guided techniques represented the gold standard for the localization and removal of non-palpable breast lesions for more than a century. Numerous disadvantages of the procedure from a patient as well as a surgeon perspective have put this standard into question for almost two decades. The introduction of intraoperative ultrasound and probe-guided technologies provided new options that currently intensify the discussion on replacement of WGL by more sophisticated technologies. While IOUS offers favorable rates of clear margins and re-excisions compared to WGL, its use is restricted to solid masses and requires high expertise from the surgeon. Radioactive seeds are cheap and RSL has shown equivalence to WGL with regard to successful lesion localization and removal. Its use is, however, not widely available due to radiation protection regulations in many countries. Non-wire and non-radio-guided techniques using magnetic or paramagnetic markers, radar reflectors, or radiofrequency identification tags are promising in this context and provide excellent early results compared to competing technologies. The devices, however, have a high upfront cost, although the cost-effectiveness of the whole pathway is not established. Carbon localization is a cost-effective option used in some countries. Modification of treatment standards and the introduction of new and potentially more cost-intensive technologies require solid evidence with regard to clinical effectiveness as well as patient and clinician satisfaction. The MELODY study aims to close this important knowledge gap by comparing all available localization techniques in a single prospective cohort study with regard to clinically relevant endpoints.

## Figures and Tables

**Figure 1 cancers-15-01173-f001:**
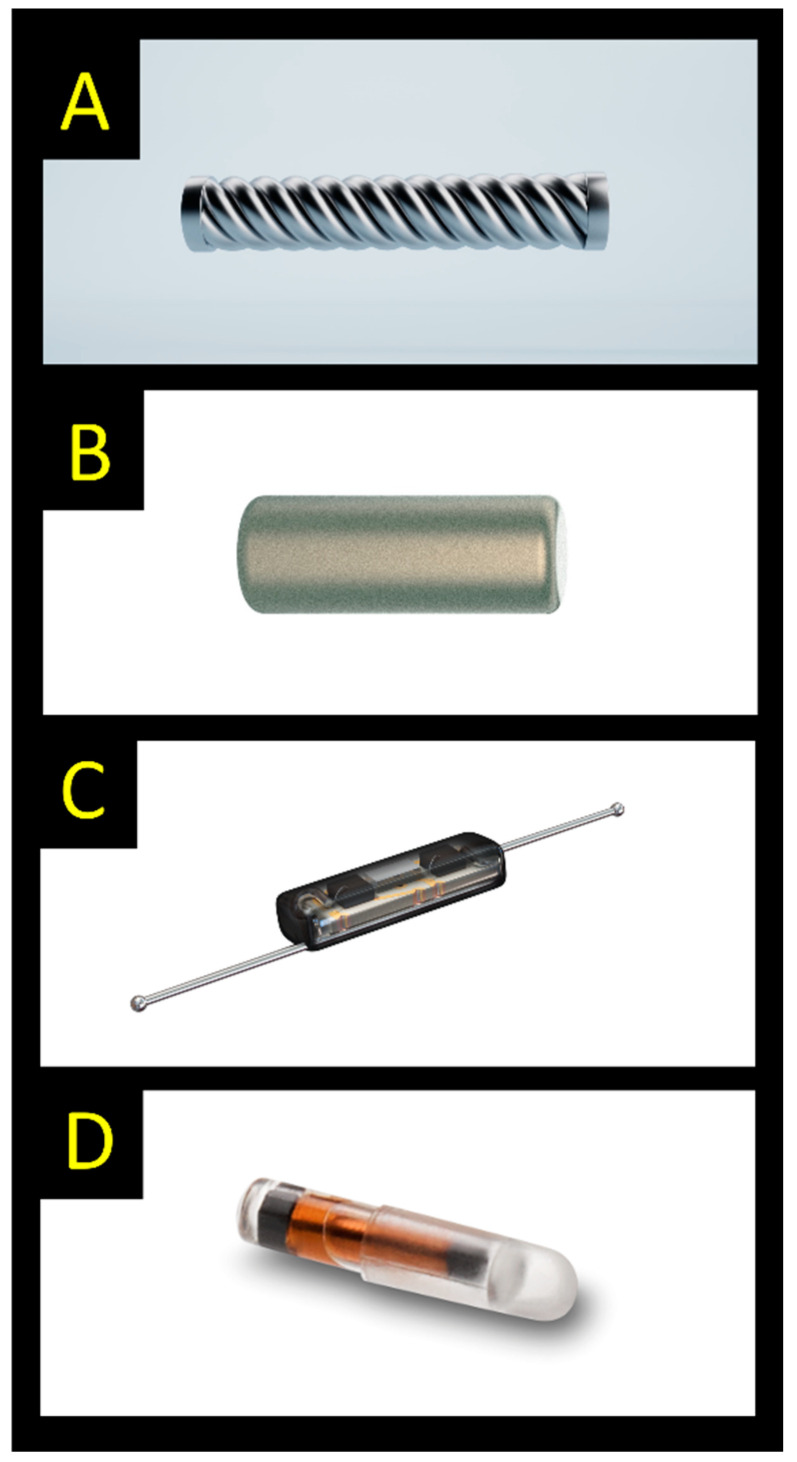
Examples of commercially available localization devices (the depicted size does not correctly compare the different markers shown): (**A**) Magseed (5 × 1 mm); (**B**) Sirius Pintuition (5 × 1.65 mm); (**C**) SAVI SCOUT (12 × 1.6 mm); (**D**) LOCalizer (11 × 2 mm) [reprinted with permission of manufacturers 2022: Endomag, Sirius Medical, Merit Medical, Hologic].

**Figure 2 cancers-15-01173-f002:**
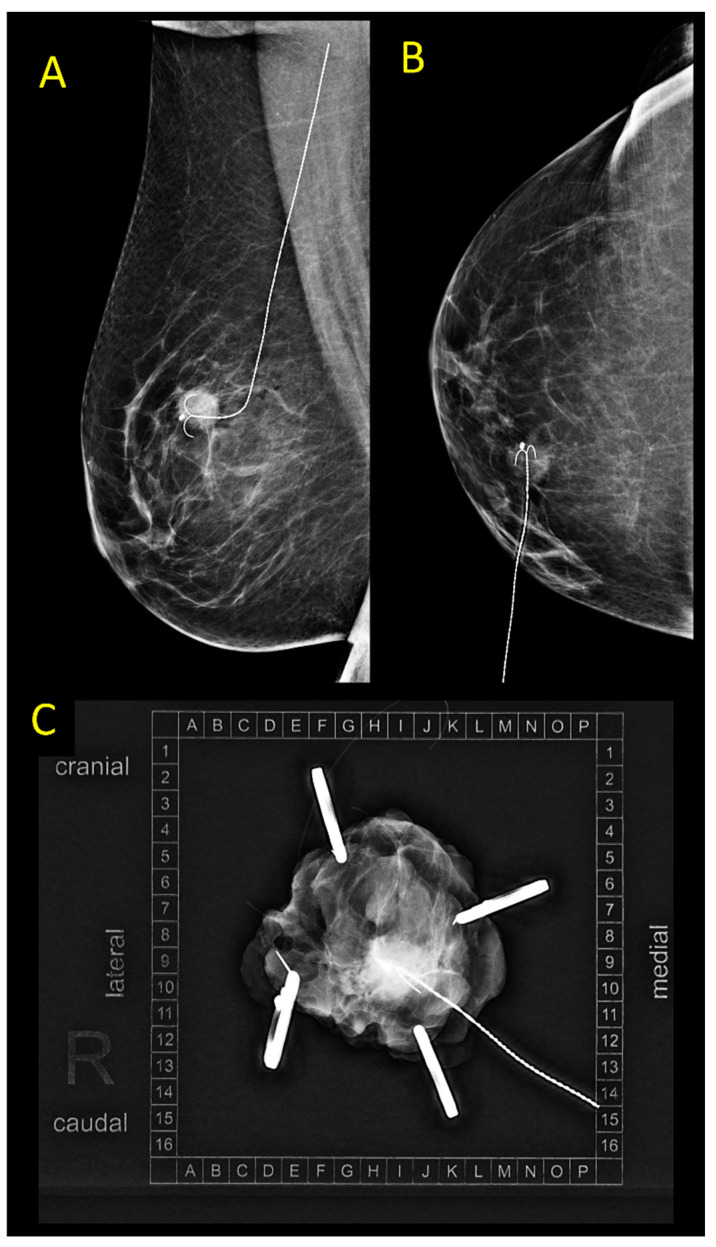
(**A**,**B**) Control mammography after ultrasound-guided wire placement in a patient with an invasive breast cancer, NST, max. size 11 mm. (**C**) Specimen mammography.

**Figure 3 cancers-15-01173-f003:**
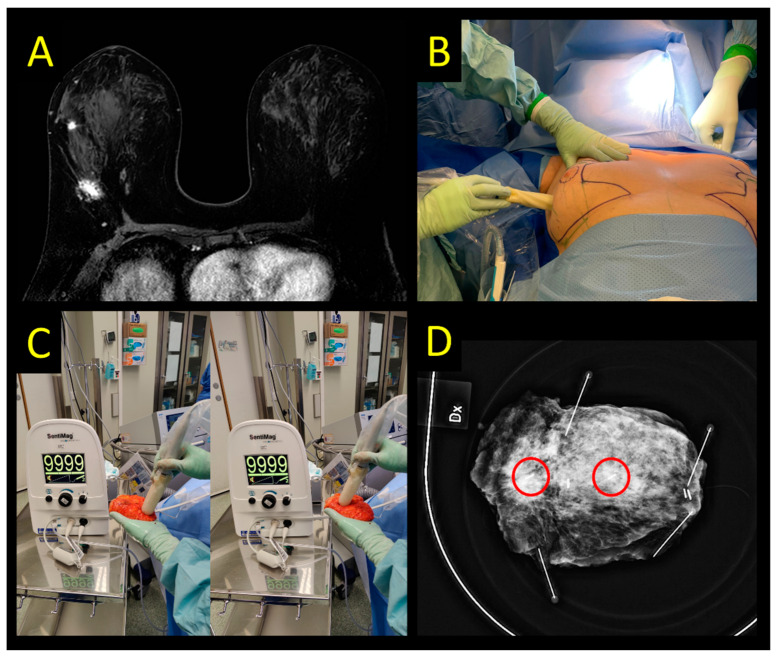
Magseed detection system. (**A**) Multicentric invasive lobular cancer (distance between lesions 4.7 cm). Each lesion is marked with a Magseed. Magtrace (SPIO) is injected between the lesions. (**B**) Transcutaneous detection with the probe. Mark the lack of skin discoloration after a deep Magtrace injection. (**C**) Ex vivo signal of the specimen. Both Magseeds have maximum signal. Observe the brown tissue staining at the SPIO injection site that does not affect specimen radiography. (**D**) Specimen radiography depicting the lesions with Magseeds (red circles highlight the position of Magseed markers).

**Figure 4 cancers-15-01173-f004:**
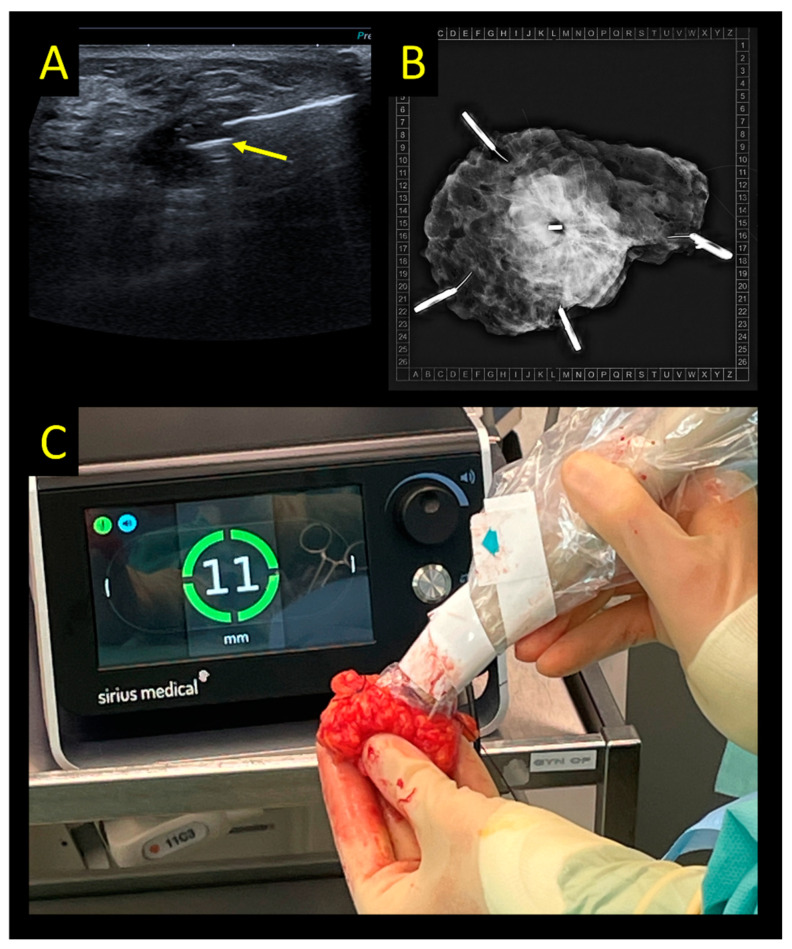
Sirius Pintuition system. (**A**) Ultrasound-guided placement of the marker (yellow arrow). (**B**) Intraoperative radiogram showing the marker in the center of the specimen. (**C**) Console used for detection in the OR showing 11 mm distance between probe tip and marker.

**Figure 5 cancers-15-01173-f005:**
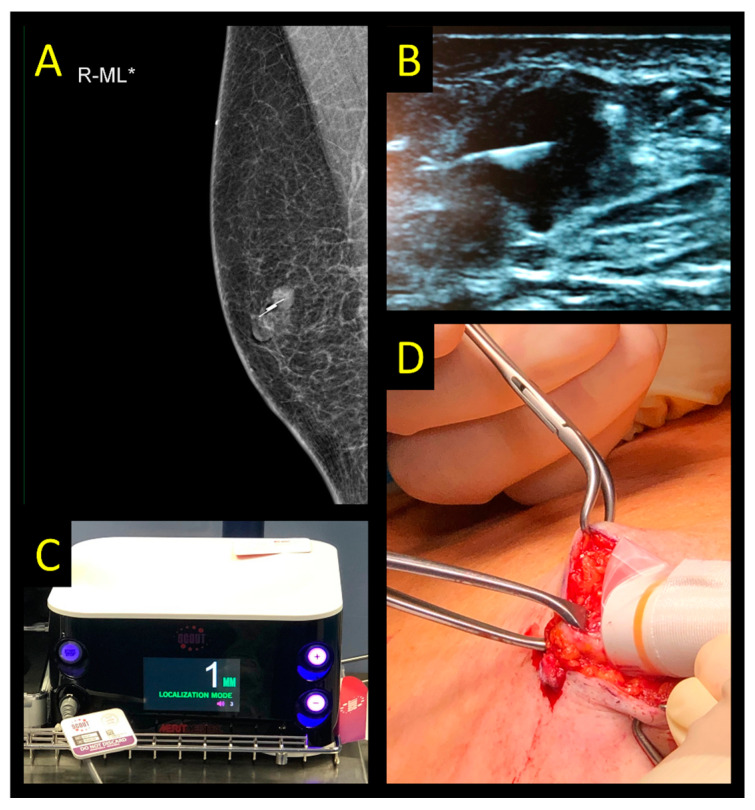
SAVI SCOUT system. (**A**) Control mammography after ultrasound-guided marker placement in a patient with an invasive breast cancer, NST, max. size: 12 mm. (**B**) Ultrasound of the marker and lesion. (**C**) Console used for detection in the operating room. (**D**) Intraoperative use of radar probe to guide excision.

**Figure 6 cancers-15-01173-f006:**
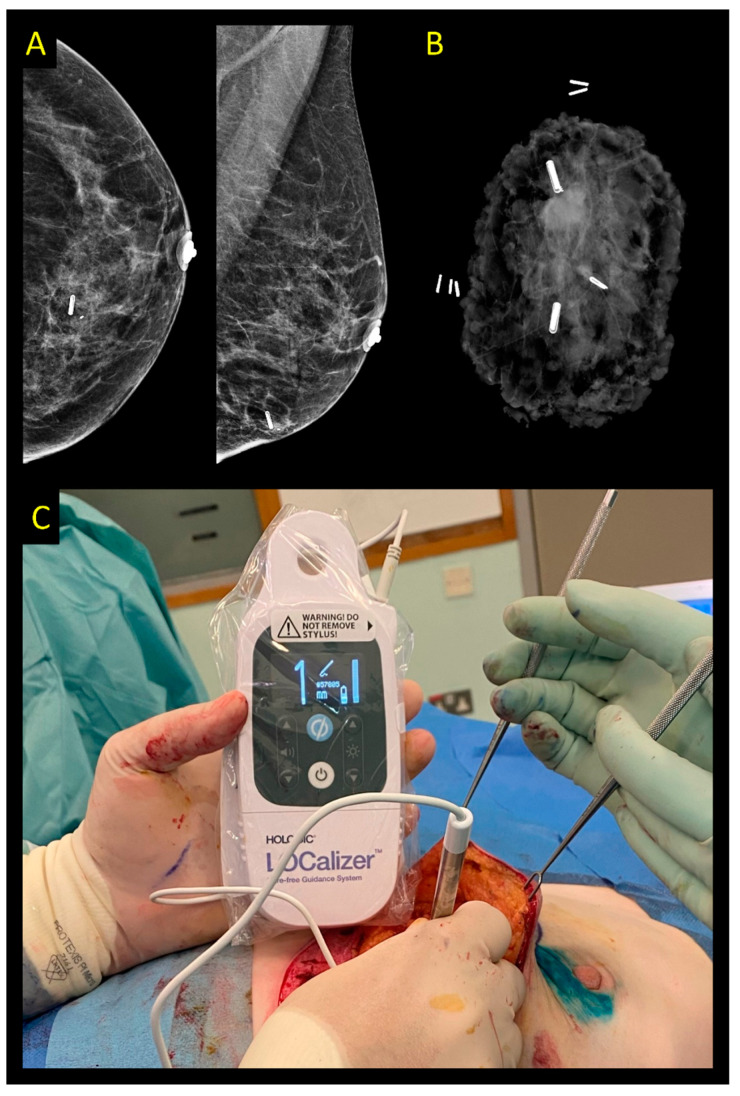
LOCalizer system. (**A**) Control mammography after ultrasound-guided placement of two RFID markers (one of them is near the thoracic wall and therefore not visible on the mammogram). (**B**) Specimen radiogram confirming the excision of both markers and the lesion. (**C**) Intraoperative use of the radiofrequency probe to guide excision.

**Figure 7 cancers-15-01173-f007:**
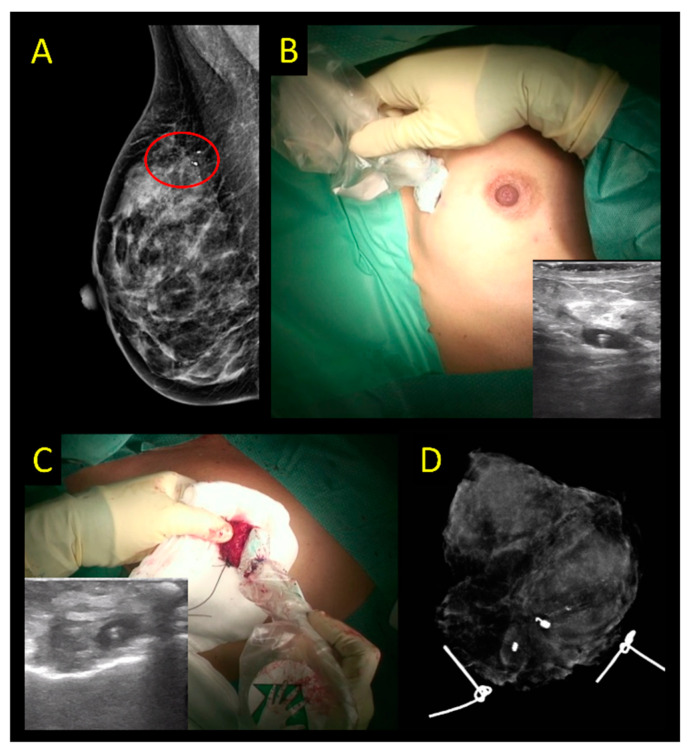
Ultrasound-guided excision of ductal carcinoma in situ with a preoperative placement of an US-visible marker. (**A**) Area of microcalcifications surrounding the US-visible marker seen on preoperative imaging. (**B**) Assessing marker before incision with IOUS with US-visible marker. (**C**) Specimen ultrasound after excision of the clip-marked area confirming marker removal. (**D**) Specimen radiograph to assess microcalcifications excised.

**Figure 8 cancers-15-01173-f008:**
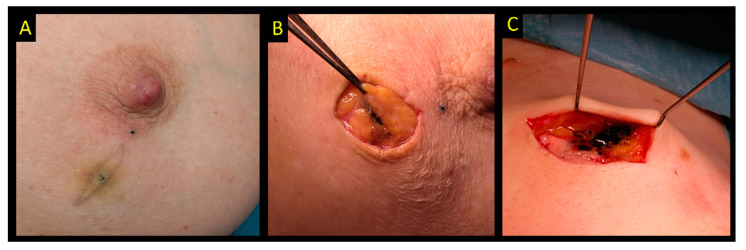
Carbon ink localization. (**A**) Intentional skin tattoo to mark the exact position of the lesion. (**B**,**C**) Intraoperative photos showing carbon ink in the tissue that will guide excision.

**Figure 9 cancers-15-01173-f009:**
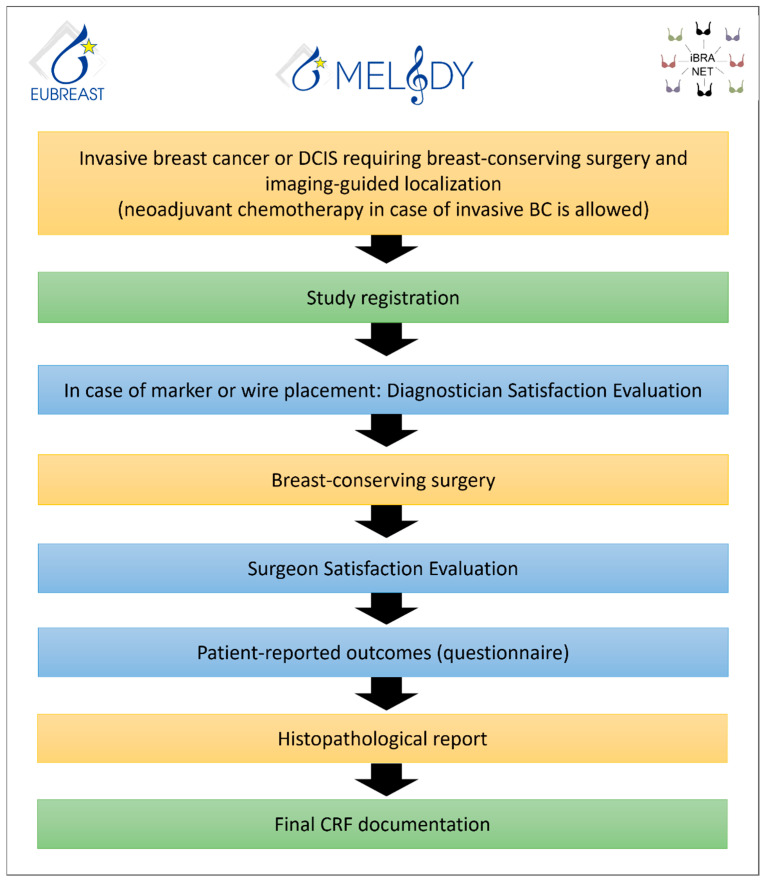
MELODY flow chart.

**Table 1 cancers-15-01173-t001:** Comparison of different localization methods regarding oncological outcomes.

	Successful Excision	Positive Margins ^1^	Re-Operation Rate	Data Quality
Wire-guided localization (WGL)	99% [9,12]	15–21% [9,10,12,14]	14–19% [9,10]	High; Meta-analyses of RCTs available (LoE 1a)
Radioactive seed localization (RSL)	100% [9]	12–13% [9,10]	10–15% [9,10]	High; Meta-analyses of RCTs available (LoE 1a)
Radio-guided Occult Lesion Localization (ROLL)	99.5% [9]	12–17% [9,10]	9–10% [9,10]	High; Meta-analyses of RCTs available (LoE 1a)
Magseed	99.8% [12]	13.3% [12]	12% [12]	Large cohort studies [12], no RCTs (LoE 2b)
Sirius Pintuition	100% [15]	8% [15]	4% [15]	Small cohort studies, one small RCT ^3^ [15] (LoE 2b)
MOLLI	100% [16]	0% [16]	0% [16]	Small phase I cohort study (LoE 4)
TAKUMI	100% [17]	7.3% [17]	4.9% [17]	Small cohort study (LoE 4)
SAVI SCOUT	99.64% [4]	n.d.	12.8% [4]	Systemic review and pooled analysis [4] (LoE 2b)
LOCalizer	99.9% [18]	n.d.	13.9% [18]	Systemic review and pooled analysis [18] (LoE 2b)
EnVisio	n.d.	n.d.	n.d.	Case report [19] (LoE 5)
Intraoperative ultrasound (IOUS)	100% [8] ^2^	5% [8,10,11] ^2^	5–7% [8,10] ^2^	High; Meta-analyses of RCTs available (LoE 1a) ^2^
Carbon	79.0–99.1% [20,21,22,23,24]	75.0–96.4% [21,22,25]	7.1% [25]	Cohort studies, no RCTs (LoE 4)

^1^ Positive margins were defined differently across studies; whenever possible, positive margin was defined as no tumor on ink. ^2^ Patients in RCTs on IOUS had ultrasound-visible lesions; therefore, the patient collective might be different from those in studies on other localization methods ^3^ The RCT studied MaMaLoc; the technology was further developed and is now available as Sirius Pintuition.

**Table 2 cancers-15-01173-t002:** Comparison of different localization methods used in breast cancer patients undergoing breast conserving surgery (modified after: [26].

	Advantages	Disadvantages
Wire-guided localization (WGL)	Well-established Cost-effective Marker placement under radiographic, ultrasound or MRI guidance possible → suitable for localization of lesions visible only upon mammography (e.g., microcalcifications) or MRI Control mammogram or MRI after wire placement possible Reposition in case of some wires possible	Scheduling issues: the wire needs to be placed on the day of surgery or the day before Wire dislocation possible Patient discomfort
Radioactive seed localization (RSL)	Well-established Scheduling flexibility: localization can be performed several days/weeks before surgery or—in case of neoadjuvant therapy—before start of treatment Marker placement under radiographic or ultrasound guidance possible → suitable for localization of lesions visible only upon mammography (e.g., microcalcifications) Control mammogram after marker placement possible Can be combined with isotope-based sentinel node biopsy	Procedure not authorized in some countries, requires complex radiation safety procedures Radiation exposure to patient and staff Invasive procedure for marker placement necessary In case of marker placement before neoadjuvant therapy signal loss possible in case of longer than planned duration of therapy Reposition after placement not possible Radiation safety concerns regarding MRI-guided localization (Geiger counter is MRI unsafe and cannot be used in case of seed loss in Zone IV) Very low risk of seed rupture or transection, resulting in emergency treatment with iodine to saturate and safeguard the thyroid gland in case of ^125^I
Radio-guided Occult Lesion Localization (ROLL)	Well-established Marker placement under radiographic, ultrasound or MRI guidance possible → suitable for localization of lesions visible only upon mammography (e.g., microcalcifications) or MRI	Scheduling issues: procedure needs to be performed on the day of surgery or the day before Radiation safety procedures required Potential radiation exposure to patient and staff Invasive preoperative procedure necessary Reposition after placement not possible Control mammogram not possible unless contrast also given
Magnetic and paramagnetic localization Commercially available systems: Magseed (Endomag) Sirius Pintuition (formerly known as MaMaLoc; Sirius Medical) MOLLI (MOLLI Surgical) TAKUMI/Guiding-marker system (Hakko)	No radioactivity involved Marker placement under radiographic or ultrasound guidance possible → suitable for localization of lesions visible only upon mammography (e.g., microcalcifications) Scheduling flexibility: localization can be performed several days/weeks before surgery or—in case of neoadjuvant therapy—before start of treatment No decrease of signal over time → reliable detectability in case of longer than planned neoadjuvant therapy Control mammogram after marker placement possible Can be combined with magnetic tracer for sentinel node biopsy	Concerns regarding use in patients with pacemakers and implantable defibrillators Standard metal surgical tools may lead to interference during measurement Large MRI artifacts Not suitable for lesions visible only upon MRI Higher device cost Adequate localization may be limited in case of a large distance between marker and detection probe Reposition after placement not possible
Radar reflector-based localization Commercially available systems: SAVI SCOUT (Merit Medical)	No radioactivity involved Minimal MRI artifact Marker placement under radiographic or ultrasound guidance possible → suitable for localization of lesions visible only upon mammography (e.g., microcalcifications) Scheduling flexibility: localization can be performed several days/weeks before surgery or—in case of neoadjuvant therapy—before start of treatment No decrease of signal over time → reliable detectability in case of longer than planned neoadjuvant therapy Control mammogram after marker placement possible	Potential signal interference with lights in the operating theatre Small MRI artifacts Not suitable for lesions visible only upon MRI Higher device cost Adequate localization may be limited in case of a large distance between marker and detection probe Reposition after placement not possible
Radiofrequency identification tags (RFID) Commercially available systems: LOCalizer (HOLOGIC) EnVisio (Elucent Medical)	No radioactivity involved Scheduling flexibility: localization can be performed several days/weeks before surgery or—in case of neoadjuvant therapy—before start of treatment Marker placement under radiographic or ultrasound guidance possible → suitable for localization of lesions visible only upon mammography (e.g., microcalcifications) No decrease of signal over time → reliable detectability in case of longer than planned neoadjuvant therapy Unique tag number → differentiation between tags possible Control mammogram after marker placement possible	Concerns regarding use in patients with pacemakers and implantable defibrillators MRI artifacts Not suitable for lesions visible only upon MRI Higher device cost Adequate localization may be limited in case of a large distance between marker and detection probe Reposition after placement not possible
Intraoperative ultrasound (IOUS)	Direct visualization during surgery No radioactivity involved Patient friendly (non-invasive) No preoperative invasive procedure necessary → scheduling flexibility Specimen sonography is performed immediately after tissue removal → no time loss due to specimen transport Specimen sonography performed in the operating room → exact and reliable topographic localization of close margins for immediate re-excision Relatively low cost	Surgeon needs to be experienced in breast ultrasound, otherwise radiologist’s presence in the operating theatre necessary Learning curve Useful only for lesions with good sonographic visibility Not suitable for lesions visible only upon mammography (e.g., microcalcifications) or MRI Use in the neoadjuvant setting limited in case of complete remission due to low sonographic visibility of some tissue markers Ultrasound machine must be available in the operating theatre during surgery Some ultrasound machines available in operating theatres are unsuitable for breast ultrasound (frequency, transducer type) or of a much lower quality than machines in the diagnostics department Radiogram showing lesion and marker not possible
Carbon	No radioactivity involved Low cost Scheduling flexibility: localization can be performed several days/weeks before surgery or—in case of neoadjuvant therapy—before start of treatment Marker placement under radiographic or ultrasound guidance possible No MRI artifacts	Marker cannot be localized without surgical exploration Possible ink migration Intentional or unintentional tattooing of skin Reposition after placement not possible Control mammogram not possible

**Table 3 cancers-15-01173-t003:** The MELODY study: Inclusion and exclusion criteria.

Inclusion Criteria	Exclusion Criteria
Signed informed consent form Malignant breast lesion requiring breast-conserving surgery and imaging-guided localization (either DCIS or invasive breast cancer; multiple or bilateral lesions and the use of neoadjuvant chemotherapy are allowed) Planned surgical removal of the lesion using one or more of the following imaging-guided localization techniques: ○ Wire-guided localization; ○ Intraoperative ultrasound; ○ Magnetic localization; ○ Radioactive seed localization; ○ Radio-guided Occult Lesion Localization (ROLL); ○ Radar localization; ○ Radiofrequency identification (RFID) tag localization; ○ Ink/carbon localization. Female/male patients ≥ 18 years old	Patients not suitable for surgical treatment Patients requiring mastectomy as first surgery Surgical removal without imaging-guided localization

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
