# Peer review of "Localization Techniques for Non-Palpable Breast Lesions: Current Status, Knowledge Gaps, and Rationale for the MELODY Study (EUBREAST-4/iBRA-NET, NCT 05559411)"

_cancers, 2023, doi:10.3390/cancers15041173_

Round 1

Reviewer 1 Report

The authors precisely and concisely present the various localization techniques used for detecting non-palpable breast lesions. However, the authors should address the following minor issues:

1) The authors should insert the citation in line no. 187.

2) In line no. 400 (citation no. 55), the authors mentioned that n=82. However total no. of participants seems to be 87 in the original article.

Author Response

Both corrections implemented. Thank you!

Reviewer 2 Report

Authors present a review article to present the background of recently initiated MELODY study (MEthods for LOcalization of Different 123 types of breast lesions) of The European Breast Cancer Research Association of Surgical Trialists (EUBREAST), which enrolls breast cancer patients undergoing breast-conserving surgery using imaging-guided localization.

Manuscript is well written and shows the scope of current knowledge on the subject, however it should be clearly defined what kind of review is this - narrative review? scoping review? Were PRISMA criteria used? How many studies were included in the study.

What is the ration for this study? MELODY is an ongoing study with all approvals; there are AGO Recommendations from the same authors which thematize more or less the same subject? 

I suggest to include and cite:

Pan B, Xu Y, Zhou Y, Yao R, Zhou X, Xu Y, Ren X, Xiao M, Zhu Q, Kong L, Mao F, Lin Y, Zhang X, Shen S, Sun Q. Long-term survival of screen-detected synchronous and metachronous bilateral non-palpable breast cancer among Chinese women: a hospital-based study (2003-2017). Breast Cancer Res Treat. 2022 Nov;196(2):409-422. doi: 10.1007/s10549-022-06747-5. Epub 2022 Sep 27. PMID: 36166112; PMCID: PMC9581860.

D'Angelo A, Trombadori CML, Caprini F, Lo Cicero S, Longo V, Ferrara F, Palma S, Conti M, Franco A, Scardina L, D'Archi S, Belli P, Manfredi R. Efficacy and Accuracy of Using Magnetic Seed for Preoperative Non-Palpable Breast Lesions Localization: Our Experience with Magseed. Curr Oncol. 2022 Nov 7;29(11):8468-8474. doi: 10.3390/curroncol29110667. PMID: 36354727; PMCID: PMC9689792.  

Author Response

Thank you for these comments. We have specified the nature of our review and included one of the two proposed citations. It was, however, not clear why we should cite the Chinese paper on bilateral breast cancer.

Reviewer 3 Report

Well organized work in good writen, definitly this article and Melody trial will provide pivotal evidence to the practice guideline in the management of non-palpable breast lesions. 

In the future, as been provided from 'NCCN Evidence Blocks',  based on the Melody trial, the author should provide the matrix to compare the most important categories regarding the localization techniques for non-palpable breast lesions.

Author Response

Thank you very much for these comments! We will definitely evaluate all localization techniques using NCCN Evidence Blocks, thank you for this inspiring idea.

Round 2

Reviewer 2 Report

Authors have sufficiently responded to reviewers remarks.